**Brief Communication:**

**New evidence further constraining Tibetan ice core chronologies to the Holocene**

Shugui Hou[1,2], Wangbin Zhang[2], Ling Fang[3,4], Theo M. Jenk[3,4], Shuangye Wu[5],

Hongxi Pang[2] and Margit Schwikowski[3, 4]

[1] School of Oceanography, Shanghai Jiao Tong University, Shanghai 200240, China

[2] School of Geographic and Oceanic Sciences, Nanjing University, Nanjing 210023, China.

[3] Laboratory of Environmental Chemistry, Paul Scherrer Institute, CH-5232 Villigen PSI, Switzerland.

[4] Oeschger Centre for Climate Change Research, University of Bern, Sidlerstrasse 5, CH-3012 Bern, Switzerland.

[5] Department of Geology, University of Dayton, Dayton, OH 45469, USA.

Correspondence to: Shugui Hou (shuguihou@sjtu.edu.cn)

**Abstract.** There is considerable controversy regarding the age ranges of Tibetan ice cores. The Guliya ice core was reported to reach as far back as ~760 ka (thousand years), whereas chronologies of all other Tibetan cores cover at most the Holocene. Here we present ages for two new ice cores reaching bedrock, from the Zangser Kangri (ZK) glacier in the northwestern Tibetan Plateau and the Shulenanshan (SLNS) glacier in the western Qilian Mountains. We estimated bottom ages of $8.90\pm_{0.57}^{0.56}$ ka BP and $7.46\pm_{1.46}^{1.79}$ ka BP for the ZK and SLNS ice core respectively, further constraining the time range accessible by Tibetan ice cores to the Holocene.

## 1 Introduction

Tibetan ice cores have provided a wealth of information for past climatic and environmental conditions at different time scales. However, there is still considerable controversy regarding their chronologies. Whereas the bedrock-reaching ice cores from Chongce, Puruogangri, Dunde and East Rongbuk (see Fig.1 for locations) were shown to cover at most the Holocene (Hou et al., 2018), the Guliya ice core, drilled in 1992 from the Guliya ice cap in the west Kunlun Mountains on the northwestern Tibetan Plateau (TP, Fig. 1), was reported to reach as far back as ~760 ka (Thompson et al., 1997). This would make it the oldest non-polar ice core known to date. In 2015, several new ice cores were recovered from the Guliya ice cap (Thompson et al., 2018). The updated chronology of the new Guliya ice cores (Zhong et al., 2018; 2020) is about an order of magnitude younger than the initially reported chronology of the Guliya1992 core, but still goes well beyond the Holocene. Here we present more evidence from the bottom age estimates of two new Tibetan bedrock ice cores from the Zangser Kangri and Shulenanshan glaciers (see Fig. 1 for locations) based on the accelerator mass spectrometry (AMS) $^{14}C$ measurements of the water-insoluble

organic carbon (WIOC) fraction of carbonaceous aerosols embedded in the glacier ice (Jenk et al., 2007, 2009; Uglietti et al., 2016). This recently established technique, which was validated by dating ice of known age (Uglietti et al., 2016), was also applied for establishing the chronology of the Tibetan Chongce ice cores (Hou et al., 2018).

**2 The new ice cores**

In April 2009, two ice cores were drilled to bedrock (127.78 m and 126.71 m in length) at an elevation of 6226 m asl (above sea level) from the Zangser Kangri (ZK) ice cap (34°18′5.8″N, 85°51′14.2″E; Fig. 1 and Fig. S1). The ice cores were drilled in dry holes using an electromechanical drill. The ZK ice cap is located on the northwestern TP, covering an area of 338 $km^2$ with an ice volume of 41.7 $km^3$ and snowlines at ~ 5700–5940 m asl (Shi et al., 2008). The ZK ice core borehole temperature ranges from -15.2 °C to -9.2 °C, with a basal temperature of -9.2 °C (An et al., 2016). The ZK ice cores were kept frozen and transported to the State Key Laboratory of Cryospheric Science at Lanzhou, where they were stored in a cold room (-20 °C) until being processed for analysis.

In May 2011, three bedrock-reaching ice cores were recovered from the Shulenanshan (SLNS) glacier (Fig. 1 and Fig. S2), with Core 1 (38°42'0"N, 97°15'8"E, 59.29 m in length) and Core 2 (38°42'0"N, 97°15'8"E, 59.78 m in length) drilled at an elevation of 5396 m asl and and Core 3 (38°42'19"N, 97°15'59"E, 81.05 m in length) at 5367 m asl respectively (Fig. S2). The SLNS glacier is located in the western Qilian Mountains, where the first Tibetan ice core reaching bedrock, the Dunde ice core (38°6'N, 96°24'E, 5325 m asl), was recovered (Thompson et al., 1989). The distance between the SLNS and the Dunde drilling sites is about 100 km (Fig. 1). The SLNS

glacier covers an area of 589 km$^2$ with an ice volume of 33.3 km$^3$ and snowline at ~

4800 m asl (Shi et al., 2008). The SLNS glacier temperature ranges from -5.6 °C to -

9.8 °C, with a basal temperature of -8.2 °C (Liu et al., 2009). The SLNS ice cores

were kept frozen from the time of drilling to final processing, being stored in a cold

70    room (-20°C) at the State Key Laboratory of Cryospheric Science at Lanzhou.

**3 Micro-radiocarbon dating**

Eight samples from the 127.78 m ZK ice core and seven samples from the 81.05 m

SLNS ice core were micro-radiocarbon dated with accelerator mass spectrometry at

75    the Laboratory for the Analysis of Radiocarbon with AMS (LARA) of the University

of Bern, using carbonaceous aerosol particles contained in the ice (Tables S1 and S2).

Details about sample preparation procedures and analytical methods can be found in

previous publications (Jenk et al., 2007; 2009; Uglietti et al., 2016; Hou et al., 2018).

In brief, for decontamination, ~5 mm outer layer was removed from the ice core

80    samples in a -20°C cold room and the remaining core samples were rinsed with ultra-

pure water in a class 100 laminar flow box. Particles contained in the melted ice

samples were filtered onto freshly preheated quartz fiber filters (Pallflex Tissuquartz,

2500QAO-UP). The filters were then heated at 340 °C for 10 min and at 650 °C for

12 min in a thermal-optical carbon analyzer (Model4L, Sunset Laboratory Inc., USA)

85    to combust and separate the water-insoluble organic carbon (WIOC), separated from

the elemental carbon (EC) fraction. For dating, the radiocarbon ($^{14}$C) in the resulting

$CO_2$ was measured by the Mini Carbon Dating System (MICADAS, 200 kV compact

AMS), equipped with a gas ion source for $^{14}$C analysis. The average overall

procedural blank used for correction of the AMS F$^{14}$C results was 1.26±0.59 μg

90    carbon (n = 115) with a F$^{14}$C of 0.69±0.15 (n = 76). Conventional $^{14}$C ages were

calibrated using OxCal v4.3 software with the IntCal13 calibration curve (Ramsey and Lee, 2013; Reimer et al., 2013).

**4 Results**

 **4.1 The depth-age relationship of the ZK ice core**

The age probability distributions of the ZK ice core are shown in Fig. S3, and the results are given in Table S1. The age probability distributions show a mostly monotonic increase in age with depth, following the trend of radioactive decay. The deepest (oldest) sample (ZK-8) was collected from the section of the ZK ice core close to the bedrock, giving an age of 8.75±0.24 ka cal BP (before present, i.e., 1950 AD). This is the oldest WIOC $^{14}$C age ever determined absolutely for Tibetan ice cores (Hou et al., 2018).

It should be noted that this $^{14}$C derived age represents the average age of the entire ice core sample section. Depending on the sampling resolution (i.e. sample length along the ice core axis), the bottom of the sample can thus be significantly older than the $^{14}$C determined average age, particularly for sections close to bedrock which experienced strong thinning by ice flow. Accordingly, the $^{14}$C age for ZK-8 only provides a lower age limit for the very bottom of the ZK-8 sample. For an estimate of the ice age close to bedrock, a modeling approach is required. We used the $^{14}$C ages and the $\beta$-activity horizon (An et al., 2016) to establish a continuous depth-age relationship of the ZK ice core (Fig. 2) by applying COPRA (COnstruction of Proxy Record from Age models), a Monte Carlo-based age modeling software (Breitenbach et al., 2012). This method was used before to establish the depth-age scale of the Mt. Ortles ice core extracted from the summit of Alto dell'Ortles in the Italian Alps (Gabrielli et al., 2016). The COPRA method can account for potential changes in accumulation and/or

strain rate, and provides an objective uncertainty estimate for each depth based on the density of dating horizons and their individual uncertainty. Applying this method, we estimated a bottom age of $8.90\pm_{0.57}^{0.56}$ ka BP. In addition, we established two alternative depth-age relationships of the ZK ice core by excluding either ZK-1 or ZK-2,

resulting in similar bottom age estimates (Fig. S4).

**4.2 The depth-age relationship of the SLNS ice core**

All ages show an increase with depth following the function of radioactive decay (Fig. 3). The age probability distributions of the SLNS ice core are shown in Fig. S5, and

the results are given in Table S2. Although being 2.1 m apart, samples SLNS-5 and SLNS-6 yield similar age distributions which nevertheless allow assuming the true age to be different by as much as 1.8 ka (1σ range). However, a potential shift to higher accumulation rates for this specific time interval cannot be excluded, and would be consistent with the findings of Herren et al. (2013) in the Tsambagarav ice

core from the Mongolian Altai observing such a shift at ~6 ka BP . From the Monte Carlo simulations of the SLNS ice core (Fig. 3), we got a bottom age estimate of $7.46\pm_{1.46}^{1.79}$ ka BP. When fitting all $^{14}$C ages with a simple exponential regression model, it gave a similar modeled age of $7.30\pm0.52$ ka BP at the ice-bedrock contact (Fig. 3).


**5 Discussion**

**5.1 The implication of the bottom age of ZK ice core**

The ages at the ice-bedrock contact were estimated to be $8.3\pm_{3.6}^{6.2}$ ka BP and $9.0\pm_{3.6}^{7.9}$ ka BP for the Chongce 216.6 m and 135.8 m ice cores respectively (Hou et al., 2018).

Our estimate of the ZK ice core bottom age of $8.90\pm_{0.57}^{0.56}$ ka BP is very close to the

older bottom age estimates of the Chongce ice cores. These bottom age estimates are much younger than the luminescence age of 42±4 ka BP for the basal sediment collected from the bottom of the Chongce ice core, which was regarded as an upper limit of the Chongce ice core bottom age (Zhang et al., 2018).

Although the ZK, Chongce and Guliya ice cores were all retrieved from the northwestern TP, their chronologies are significantly different. Thompson et al. (1997) suggested the top 266 m of the Guliya1992 ice core covers the past 110 ka based on matching the Guliya $\delta^{18}O$ record with the GISP2 ice core $CH_4$ record. They believed the Guliya 1992 ice core was older than 500 ka BP below the depth of 290

m, and up to ~760 ka BP at the ice-bedrock contact, primarily based on the $^{36}Cl$ measurements. These age estimates are near two orders of magnitudes older than all the other Tibetan ice cores. In 2015, a new Guliya ice core to bedrock (309.73 m in length) was recovered adjacent to the drilling site in 1992. At the same time, three ice cores to bedrock (50.72 m, 51.38 m and 50.86 m in length) were retrieved from the

Guliya ice cap summit (35°17′ N, 81°29′ E; ~ 6700 m asl) (Thompson et al., 2018). So far, only the 50.86 m Guliya2015 summit core has some limited published information on its chronology. Zhong et al. (2018) indicated that the 50.86 m Guliya2015 summit core is ~20 ka BP at the depth of 41.10 - 41.84 m and ~30 ka BP at 49.51 - 49.90 m based on matching its $\delta^{18}O$ profile with that of the Guliya1992 ice

core. Later the same research team refined those two ages from ~20 ka BP to ~ 4 – 4.5 ka BP and from ~30 ka BP to ~ 15 ka BP at the same depths(Zhong et al., 2020). Extrapolating from the two age points, Hou et al. (2019) estimated the basal ages of the Guliya2015 summit core to be 76.6 ka BP (or 91.7 ka BP after refinement) at 0.01 m w.e. above bedrock, and 48.6 ka BP (or 29.0 ka BP after refinement) at 0.20 m w.e.

above bedrock (Fig. S6).

Several studies have already raised questions about the accuracy of the Guliya ice

core chronology (Cheng et al., 2012; Hou et al., 2018; 2019; Tian et al., 2019). Cheng

et al. (2012) argued that the 110 ka time scale needs to be compressed by a factor of

two in order to reconcile the difference of the $\delta^{18}O$ variations between the Guliya ice

core and the Kesang stalagmite records (see Fig. 1 for location). Tian et al. (2019)

provided the first radiometric $^{81}Kr$ dating results for ice samples collected at the

outlets of the Guliya ice cap, yielding upper age limits in the range of 15–74 ka (90%

confidence level). Moreover, Hou et al. (2019) found a high degree of consistency

between the depth - $\delta^{18}O$ profiles of the Guliya and Chongce ice cores, and argued

that the Guliya ice core might cover a similar age range as the Chongce core. Several

factors could lead to difference in ice core age, such as the difference in annual

precipitation, base topography, ice cap dynamics and drilling locations. However, the

new estimates of the bottom ages of the ZK ice core call for further investigation in

the significant difference between the Guliya and all other Tibetan ice core

chronologies.

**5.2 The implication of the bottom age of SLNS ice core**

Both the SLNS and Dunde ice cores were recovered from the western Qilian

Mountains (Fig. 1). In 1987, the Dunde ice cores (139.8 m, 136.6 m and 138.4 m in

length, respectively) were drilled to bedrock from the Dunde ice cap (38°6′N,

96°24′E, 5325 m asl). The 139.8 m Dunde ice core was previously dated to be 40 ka

BP at the depth of 5 m above the bedrock, and potentially >100 ka BP at the ice-

bedrock contact (Thompson et al., 1989). Later, a single $^{14}C$ age of 6.24±0.33 ka BP

was determined for a sample collected close to bedrock (exact distance above the

bedrock unavailable) (Thompson et al., 2005). The $^{14}C$ cal age of 6.62±0.82 ka BP

for a sample at 0.03 m above the bedrock of the SLNS ice core is in good agreement with 6.24±0.33 ka of the Dunde ice core sample, suggesting that the Dunde ice core may be of Holocene origin (Thompson et al., 2005).

**6 Conclusions**

We presented the $^{14}$C ages of two new Tibetan bottom ice cores, providing additional support for the Holocene origin of Tibetan ice cores. These results are much younger than the original chronology for the Guliya ice core, and could have a significant impact on the interpretation of climate record of the region. In order to resolve the

chronology discrepancy between the Guliya and the other Tibetan ice cores, it is necessary to explore more independent lines of evidence, especially from absolute dating techniques (e.g. $^{14}$C, $^{36}$Cl, $^{10}$Be and $^{81}$Kr) and ice core gas measurements (e.g. $CH_4$, the isotopic composition of atmospheric $O_2$). Moreover, the FAIR (Findable, Accessible, Interoperable and Reusable) sharing of the Tibetan ice core original

datasets (e.g., water isotopes, major ions, dust, gas measurements) would provide tremendous benefit for future research in this field.

**Data availability.** The $^{14}$C data of the ZK and SLNS ice cores is provided in Tables S1 and S2.


**Author contribution.** SH conceived this study, drilled the ice cores, and wrote the paper with contributions from SW, MS and TMJ. LF, TMJ and MS measured the $^{14}$C sampels. WZ prepared the figures with contributions from SH, MS and TMJ. All authors contributed to discussion of the results.


**Competing interests.** The authors declare no conflict of interest.

**Acknowledgments.** Thanks are due to many scientists, technicians, graduate students and porters, especially to Yongliang Zhang and Yaping Liu, for their great efforts in
the high elevations. This work was supported by the National Natural Science Foundation of China (91837102, 41830644 and 42021001) and the "333 Project" of the Jiangsu Province (BRA2020030).

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

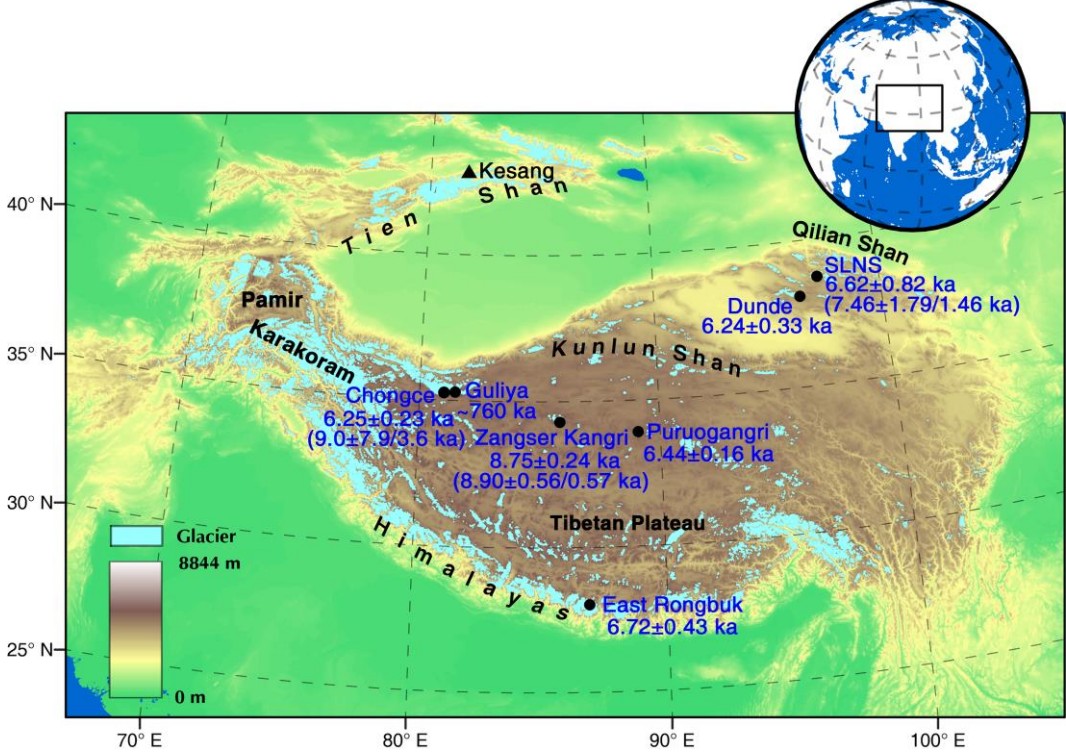


Figure 1. Map showing the locations of ice core drilling sites. The numbers for each site except Guliya are the oldest measured $^{14}$C ages, while the number inside the bracket is the estimated ice age at the ice-bedrock contact. Data of glaciers are from the Global Land Ice Measurements from Space (GLIMS, available at

http://www.glims.org). The topographic data were extracted from ETOPO1 elevations global data, available from National Oceanic and Atmospheric Administration at http://www.ngdc.noaa.gov/mgg/global/global.html.

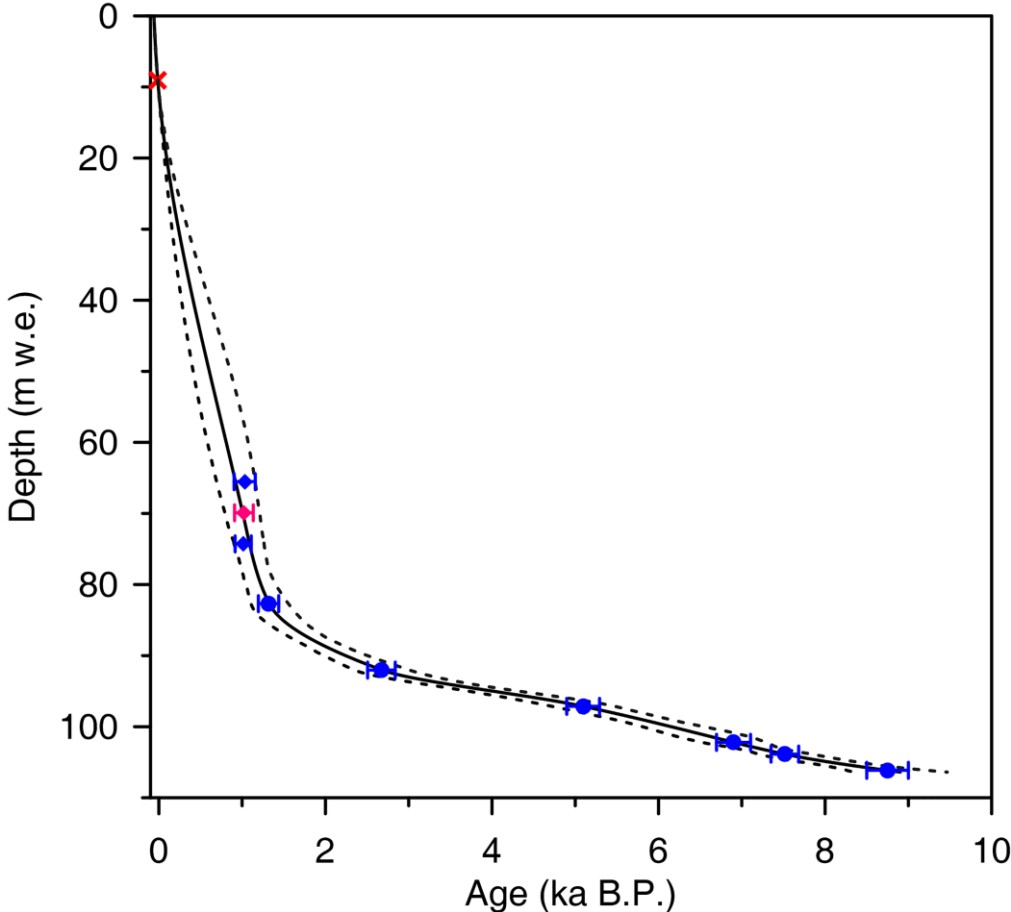

Figure 2. The age–depth relations of the ZK ice core based on 2000 Monte Carlo

simulations fitting the absolute dated age horizons. Solid black lines indicate the mean

values and dotted lines indicate the 1σ confidence interval. The red cross stands for

the reference layer of *β*-activity peak in 1963 (An et al., 2016). Blue circles show the

calibrated WIOC [14]C ages, and red dot represents the average of the ZK-1 and ZK-2

ages at their average depth. Errors bars represent the 1σ uncertainty. Note that ZK1

and ZK-2 are not included in the Monte Carlo simulations, but both are located within

the 1σ confidence envelopes.

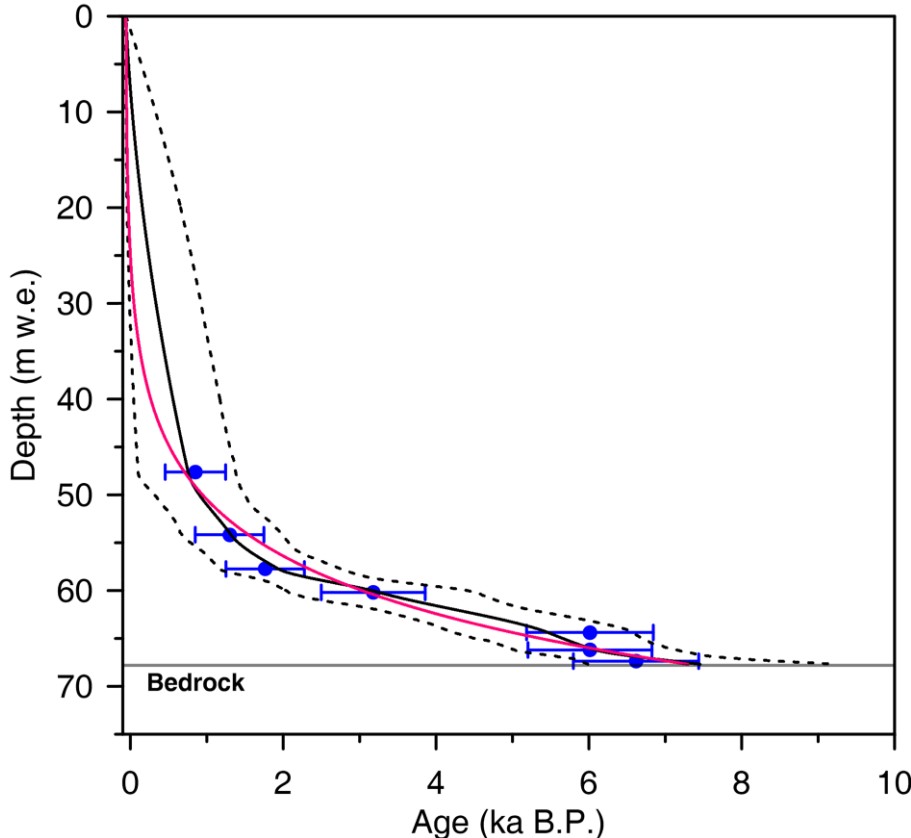

Figure 3. The age–depth relationships of the SLNS ice core based on 2000 Monte

Carlo simulations fitting the absolute dated age horizons. Solid black line indicates

model mean and dotted lines indicate the $1\sigma$ confidence interval. The blue dots stand

for the calibrated $^{14}$C ages with $1\sigma$ error bar. The red line shows the depth–age profile

modeled by an exponential regression.