# Peer review of "Brief Communication"

_The Cryosphere, 2020_

## Referee Comment (RC1) · Paul Andrew Mayewski (Referee) · 8 Sep 2020

The research presented by Hou et al.provides an alternative view concerning the bottom age of Tibetan ice cores based on radiocarbon dated ice core records from two ice cores. It is important that this alternative view be made publicly available since it departs from previous studies by an order of magnitude in age by suggesting that the bottom age is Holocene in age rather than hundreds of thousands of years, thus significantly impacting past climate reconstructions for the region.

---

## Referee Comment (RC2) · Anonymous Referee #2 · 28 Dec 2020

Hou et al. present the 14C dating results of water-insoluble organic carbon (WIOC) fraction of carbonaceous aerosols from two new ice cores reaching bedrock (i.e., the Zangser Kangri (ZK) glacier and the Shulenanshan (SLNS) glacier) in northern Tibetan Plateau. These 14C dates suggest that the bottom ages of these two ice cores are ~8.90 ±0.57 and ~7.46 ±1.79 ka, respectively. Considering the newly obtained bottom ages of these two ice cores and other bottom ages determined absolutely for other ice cores on the Tibetan Plateau, the authors therefore argue for the Holocene origin of Tibetan ice cores and challenge the reliability of chronologies of the Guliya and Dunde ice cores.

[Figure]

The dates from these two new ice cores are no doubt essential for establishing their own absolute chronologies. Based on the bottom ages of these two ice cores along with other ice cores' bottom ages to argue the Holocene origin of Tibetan ice cores sounds reasonable, but it is not a definitive conclusion, since the difference in annual precipitation, the base topography, the dynamics of ice cap and the position of drilling core, all these factors may affect the base age of ice core extracted. Using only the ages is insufficient to argue the accuracy of the original chronology of Guliya ice core. The changes in chronology may induce significant change in the proxy time series, such as $\delta$18O records. Actually, the authors have used the comparison of the $\delta$18O record from Chongce ice core and other ice cores to argue the bottom ages of Guliya and Dunde ice cores. However, the comparison seems not so successful as show in the paper by Hou et al., (2019). If the proxy time series with these corrected bottom ages could also correlate with these two new ice cores, the arguments would be robust. So I suggest authors tempering the arguments, because there is no such direct dating results from those previously reported ice cores or proxy records comparisons.

As mentioned in the manuscript and also in two published papers (Hou et al., 2018; 2019), the bottom age of Dunde ice core has been changed close to ∼6.2 ka BP and the original chronology of this ice core has been corrected in the paper by Thompson et al. (2005). I suggest the author use this as an evidence to argue the Holocene origin of Tibetan Plateau, since the estimated bottom age and the original chronology have been corrected already. The argument in lines from 190 to 195 are meaningless and should be deleted.

For the discussion section, I suggest the authors change the subtitles a little bit, such as " the implication to the bottom age of XX ice cores".

---

## Referee Comment (RC3) · Anonymous Referee #3 · 28 Jan 2021

I read with great interest the MS entitled "New evidence further constraining Tibetan ice core chronologies to the Holocene" by Shugui Hou. This is the result of a great effort of micro-radiocarbon dating in two Tibetan cores from the Zangser Kangri glacier in the northwestern Tibetan Plateau and the Shulenanshan glacier in the western Qilian Mountains. It present a much younger age for the bottom ice at these two sites, compared to what was found for the Guliya ice cap. I think this is an important point of view and it deserves to be published as a brief communication in TC. This will have a significant impact in the interpretation of climate record of the region.

---

## Author Comment (AC1) · 16 Feb 2021

Dear Prof. Paul Andrew Mayewski,

Many thanks for your positive review of our work. We agree with you that our alternative view concerning the bottom age of Tibetan ice cores may significantly impact past climate reconstructions for the region, and we look forward to new results and understandings on this fundamental issue.

Sincerely yours,

Hou Shugui, on behalf of all co-authors

---

## Author Comment (AC3) · 16 Feb 2021

Dear Referee,

Many thanks for your positive review of our work. We agree with you that our new understanding on the chronology of the Tibetan ice cores "will have a significant impact in the interpretation of climate record of the region", and we look forward to more research and results on this fundamental issue.

Sincerely yours,

Hou Shugui, on behalf of all co-authors

---

## Author Response (AR1)

Paul Andrew Mayewski (Referee)

The research presented by Hou et al. provides an alternative view concerning the bottom age of Tibetan ice cores based on radiocarbon dated ice core records from two ice cores. It is important that this alternative view be made publicly available since it departs from previous studies by an order of magnitude in age by suggesting that the bottom age is Holocene in age rather than hundreds of thousands of years, thus significantly impacting past climate reconstructions for the region.

Response:

Many thanks for the positive review of our work. We agree with you that our alternative view concerning the bottom age of Tibetan ice cores may significantly impact past climate reconstructions for the region, and we look forward to new results and understandings on this fundamental issue.

Anonymous Referee #2:

Hou et al. present the 14C dating results of water-insoluble organic carbon (WIOC) fraction of carbonaceous aerosols from two new ice cores reaching bedrock (i.e., the Zangser Kangri (ZK) glacier and the Shulenanshan (SLNS) glacier) in northern Tibetan Plateau. These 14C dates suggest that the bottom ages of these two ice cores

are ~8.90 ±0.57 and ~7.46 ±1.79 ka, respectively. Considering the newly obtained bottom ages of these two ice cores and other bottom ages determined absolutely for other ice cores on the Tibetan Plateau, the authors therefore argue for the Holocene origin of Tibetan ice cores and challenge the reliability of chronologies of the Guliya and Dunde ice cores.

The dates from these two new ice cores are no doubt essential for establishing their own absolute chronologies. Based on the bottom ages of these two ice cores along with other ice cores' bottom ages to argue the Holocene origin of Tibetan ice cores sounds reasonable, but it is not a definitive conclusion, since the difference in annual precipitation, the base topography, the dynamics of ice cap and the position of drilling core, all these factors may affect the base age of ice core extracted. Using only the ages is insufficient to argue the accuracy of the original chronology of Guliya ice core. The changes in chronology may induce significant change in the proxy time series, such as $\delta 18O$ records. Actually, the authors have used the comparison of the $\delta 18O$ record from Chongce ice core and other ice cores to argue the bottom ages of Guliya and Dunde ice cores. However, the comparison seems not so successful as show in the paper by Hou et al., (2019). If the proxy time series with these corrected bottom ages could also correlate with these two new ice cores, the arguments would be robust. So I suggest authors tempering the arguments, because there is no such direct dating results from those previously reported ice cores or proxy records comparisons.

Response:

We agree with the reviewer's comments, and have tempered the arguments in the revised text to focus more on data and less on speculations.

As mentioned in the manuscript and also in two published papers (Hou et al., 2018; 2019), the bottom age of Dunde ice core has been changed close to ~6.2 ka BP and the original chronology of this ice core has been corrected in the paper by Thompson et al. (2005). I suggest the author use this as an evidence to argue the Holocene origin of Tibetan Plateau, since the estimated bottom age and the original chronology have been corrected already. The argument in lines from 190 to 195 are meaningless and should be deleted.

Response:

We agree with the comment, and have deleted the argument in lines from 190 to 195 in the revision.

For the discussion section, I suggest the authors change the subtitles a little bit, such as "the implication to the bottom age of XX ice cores".

Response:

We agree with the comment, and have changed the subtitles accordingly in the revision.

Anonymous Referee #3:

I read with great interest the MS entitled "New evidence further constraining Tibetan ice core chronologies to the Holocene" by Shugui Hou. This is the result of a great effort of micro-radiocarbon dating in two Tibetan cores from the Zangser Kangri glacier in the northwestern Tibetan Plateau and the Shulenanshan glacier in the western Qilian Mountains. It presents a much younger age for the bottom ice at these two sites, compared to what was found for the Guliya ice cap. I think this is an important point of view and it deserves to be published as a brief communication in TC. This will have a significant impact in the interpretation of climate record of the region.

Response:

Many thanks for the positive review of our work. We agree with you that our new understanding on the chronology of the Tibetan ice cores "will have a significant impact in the interpretation of climate record of the region", and we look forward to more research and results on this fundamental issue.

---

## Author Response (AR2)

Dear Dr. Kerim Nisancioglu,

Many thanks for your time and effort in managing our contribution. I have revised the manuscript based on your comments, namely, modifications for Fig. 1 and Fig. S2 and removing the words "separated" and "respectively" in the according lines. I hope that the revised manuscript is acceptable for publication.

Sincerely yours,

Hou Shugui